# Oral angiotensin-converting enzyme inhibitor captopril protects the heart from *Porphyromonas gingivalis* LPS-induced cardiac dysfunction in mice

Kenichi Kiyomoto[1,2]☯, Ichiro Matsuo[2]☯, Kenji Suita[1], Yoshiki Ohnuki[1], Misao Ishikawa[3], Aiko Ito[4], Yasumasa Mototani[1], Michinori Tsunoda[1,2], Akinaka Morii[1,2], Megumi Nariyama[5], Yoshio Hayakawa[6], Yasuharu Amitani[7], Kazuhiro Gomi[2], Satoshi Okumura[1]*

1 Department of Physiology, Tsurumi University School of Dental Medicine, Yokohama, Japan,
2 Department of Periodontology, Tsurumi University School of Dental Medicine, Yokohama, Japan,
3 Department of Oral Anatomy, Tsurumi University School of Dental Medicine, Yokohama, Japan,
4 Department of Orthodontology, Tsurumi University School of Dental Medicine, Yokohama, Japan,
5 Department of Pediatric Dentistry, Tsurumi University School of Dental Medicine, Yokohama, Japan,
6 Department of Dental Anesthesiology, Tsurumi University School of Dental Medicine, Yokohama, Japan,
7 Department of Mathematics, Tsurumi University School of Dental Medicine, Yokohama, Japan

☯ These authors contributed equally to this work.
* okumura-s@tsurumi-u.ac.jp

**Data Availability Statement:** All relevant data are within the manuscript and its Supporting Information files.

## Abstract

Although angiotensin converting enzyme (ACE) inhibitors are considered useful for the treatment of human heart failure, some experimental failing-heart models have shown little beneficial effect of ACE inhibitors in animals with poor oral health, particularly periodontitis. In this study, we examined the effects of the ACE inhibitor captopril (Cap; 0.1 mg/mL in drinking water) on cardiac dysfunction in mice treated with *Porphyromonas gingivalis* lipopolysaccharide (PG-LPS) at a dose (0.8 mg/kg/day) equivalent to the circulating level in patients with periodontal disease. Mice were divided into four groups: 1) Control, 2) PG-LPS, 3) Cap, and 4) PG-LPS + Cap. After1 week, we evaluated cardiac function by echocardiography. The left ventricular ejection fraction was significantly decreased in PG-LPS-treated mice compared to the control (from 66 ± 1.8 to 59 ± 2.5%), while Cap ameliorated the dysfunction (63 ± 1.1%). The area of cardiac fibrosis was significantly increased (approximately 2.9-fold) and the number of apoptotic myocytes was significantly increased (approximately 5.6-fold) in the heart of PG-LPS-treated group versus the control, and these changes were suppressed by Cap. The impairment of cardiac function in PG-LPS-treated mice was associated with protein kinase C δ phosphorylation (Tyr-311), leading to upregulation of NADPH oxidase 4 and xanthine oxidase, and calmodulin kinase II phosphorylation (Thr-286) with increased phospholamban phosphorylation (Thr-17). These changes were also suppressed by Cap. Our results suggest that the renin-angiotensin system might play an important role in the development of cardiac diseases induced by PG-LPS.

**Funding:** This study was supported by the Japan Society for the Promotion of Science (JSPS) KAKENHI Grant (22K21003 to IM, 20K10305 to KS, 22K10304 to YO, 19K24109 and 21K17171 to AI, 22K10255 to MN, and 21K10242 to SO); Research. The founders had no role in study design, data collection and analysis, decision to publish, or preparation of the manuscript.

**Competing interests:** The authors have declared that no competing interests exist.

## Introduction

Angiotensin II (Ang II) is one of the longest-known peptide hormones and is a major regulator of the renin-angiotensin system (RAS), which maintains the blood pressure through vasoconstriction and modulation of salt and water homeostasis. In addition to the systemic RAS, local systems have been described in the kidney, the heart and the brain [1]. These are responsible for the much higher Ang II concentrations observed in these organs in comparison to the plasma concentration of Ang II, and contribute to the pathogenesis of several aging-associated human diseases, including hypertension, myocardial infarction, congestive heart failure, stroke, atrial fibrillation and coronary artery disease [1]. Large population studies have clearly demonstrated that ACE inhibitors are effective in preventing or inducing regression of some of these diseases in humans and animals [2].

Oral frailty has recently been suggested as a novel construct, defined as a decrease in oral function with a coexisting decline in cognitive and physical function, and could be linked to aging via changes in nutritional status (dentition impact on the nutritional status), biological pathways associated with periodontal disease, and psychological factors such a lack of self-esteem and decreased quality of life. We recently demonstrated that persistent subclinical exposure to *Porphyromonas gingivalis* lipopolysaccharide (PG-LPS) induces myocyte apoptosis and fibrosis in cardiac muscle in mice with activation of the cyclic AMP (cAMP)/protein kinase A and cAMP/calmodulin kinase II (CaMKII) pathways in mice [3]. We also showed that activation of $\beta_1$-adrenergic receptor ($\beta_1$-AR) signaling, which is involved in cardiac myocyte apoptosis, is important for the development of masseter muscle fibrosis and myocyte apoptosis via activation of CaMKII, leading to phospholamban (PLB) phosphorylation on Ser-16 and Thr-17 [4]. More importantly, activation of both the sympathetic nervous system (SNS) and the RAS was demonstrated to be associated with cardiovascular disease, including heart failure and thus worldwide standard guidelines for treating heart failure involve inhibition of the chronically enhanced SNS activity with $\beta$-blockers [5] and RAS inhibitors [2].

The aim of this study was to examine the effects of the ACE inhibitor captopril (Cap) on PG-LPS-induced cardiac dysfunction in mice by evaluating cardiac function, histology, and signal transduction in the heart of PG-LPS-administered mice treated or not treated with Cap.

## Materials and methods

### Mice and experimental protocol

All experiments were performed on male 12-week-old C57BL/6 mice obtained from CLEA Japan (Tokyo, Japan). Mice were group-housed at 23˚C under a 12–12 light/dark cycle with lights on at 8:00 AM in accordance with the standard conditions for mouse studies by our group [6–8]. Both food and water were available ad libitum.

PG-LPS (#14966–71; Invivogen, San Diego, CA, USA) was dissolved in saline to prepare a 0.6 mg/ml stock solution [9], and an appropriate volume of this solution to provide the desired dose (PG-LPS: 0.8 mg/kg) was added to 0.2 mL of saline to prepare the solution for intraperitoneal (i.p.) injection (once daily for 1 week). Mice were group-housed (approximately 3 per cage) and were divided into four groups: a normal control group (Control), a PG-LPS treatment group, a Cap-only treatment group (Cap), and a PG-LPS plus Cap treatment group (PG-LPS + Cap) (**Fig 1A**). Cap (#C8856; Sigma-Aldrich, St. Louis, MO, USA) was directly dissolved in drinking water (0.1 mg/mL; freshly prepared every day) [10, 11]. Body weight (BW), food intake, and water intake were monitored throughout the 1-week experimental period (Control: $n = 6$, PG-LPS: $n = 7$, Cap: $n = 6$, PG-LPS + Cap: $n = 7$). The dose of PG-LPS used in this study is consistent with the circulating levels in patients with periodontitis, so that this

## A

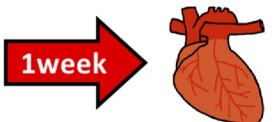

**Animals**

Mice:C57BL/6J male 12weeks (n=6~7)
· Control
· PG-LPS ( 0.8 mg/kg/day i.p. )
· Captopril (Cap)
· PG-LPS+Cap

1week

**Analysis**

· Body weight
· Serum Ang II level (ELISA)
· Cardiac function (Echocardiography)
· Organ weight
· Cardiac fibrosis area (Masson-trichrome staining)
· Cardiac apoptosis (TUNEL staining)
· Western blotting

## B

|  | Control | PG-LPS | Cap | PG-LPS+Cap |
|---|---|---|---|---|
| n | 6 | 7 | 6 | 6 |
| Food (g/day) | 9.8 ± 0.4 | 10.7 ± 0.5 | 9.0 ± 0.2 | 9.9 ± 0.8 |
| Water (mL/day) | 4.4 ± 0.1 | 4.5 ± 0.3 | 4.0 ± 0.1 | 3.9 ± 0.2 |

## C

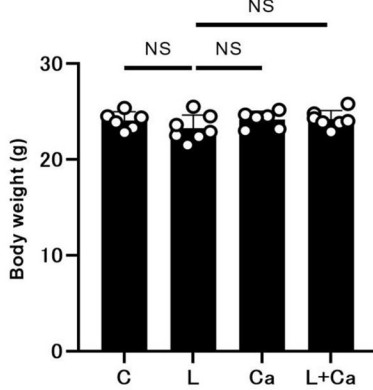

**Fig 1. Experimental procedure and consumption of food and water during chronic PG-LPS infusion in mice. (A)** Lipopolysaccharide derived from *Porphylomonas gingivalis* (PG-LPS) was administered once daily for 1 week via intraperitoneal injection (i.p.) at a dose of 0.8 mg/kg, dissolved in saline. Age-matched control mice (Control) received an identical volume of saline only. **(B)** Consumed amounts of food and water were similar among the four groups. *P* = NS vs. Control). NS, not significantly different from the Control (*P* > 0.05) by one-way ANOVA followed by the Tukey-Kramer *post hoc* test. Data shows means ± SD and open circles show individual data. **(C)** The Control (C), PG-LPS (L), Cap (Ca) and PG-LPS + Cap (L + Ca) groups showed similar body weight at 1 week after the PG-LPS infusion. NS, not significantly different from the Control (*P* > 0.05) by one-way ANOVA followed by the Tukey-Kramer *post hoc* test.

model is not a sepsis model, and indeed, no mortality was observed [9]. After the completion of treatment, mice were anesthetized with isoflurane. Blood sampling was done within 30 sec from the time of contact with the mouse. The heart, lungs and liver were excised, weighed, frozen in liquid nitrogen, and stored at -80˚C. The ratio of organ mass (mg) to tibial length (TL; mm) was used as an index of organ volume. After tissue extraction, the mice were anesthetized via a mask with isoflurane (1.0–1.5% v/v) and killed by cervical dislocation [12]. Serum angiotensin II (Ang II) levels were determined using an angiotensin II ELISA Kits (#ADI-900-204; Enzo Life Science, Farmingdale, NY, USA) in accordance with the manufacturer's instruction.

### Ethical approval

All animal experiments complied with the ARRIVE guidelines [13] and were carried out in accordance with the National Institutes of Health guide for the care and use of laboratory animals [14] and institutional guidelines. The experimental protocol was approved by the Animal Care and Use Committee of Tsurumi University (No. 29A041).

## Physiological experiments

Mice were anesthetized with isoflurane vapor (1.0–1.5% v/v) titrated to maintain the lightest anesthesia possible and echocardiographic measurements were performed by ultrasonography (TUS-A300, Toshiba, Tokyo, Japan) as described previously [15].

All LV dimensions are presented as the average of four consecutive selected beats. Heart rate (HR) was determined from the cardiac cycles recorded on the M-mode tracing, using at least three consecutive beats. The other parameters were calculated from M-mode-derived LV dimensions using the Teichholz formula [16]:

• EDV = $(7 \times \text{LVIDd}^3/1000) / (2.4 + (\text{LVIDd}/10))$ and ESV = $(7 \times \text{LVIDs}^3/1000) / (2.4 + (\text{LVIDd}/10))$ (mL)

EDV (mL): left ventricular end-diastolic volume

ESV (mL): left ventricular end-systolic volume

LVIDd (mm): left ventricular internal dimension at end-diastole

LVIDs (mm): left ventricular internal dimension at end-systole

• Stroke volume (SV) = EDV- ESV (mL)

• Cardiac output (CO) = HR x SV (ml/min)

• Left ventricular ejection fraction (EF) = 100 x SV / EDV (%)

• Left ventricular fractional shortening (%FS) = 100 x (LVIDd—LVIDs) / LVIDd (%)

All LV dimensions calculated using Teichholz formula in wild-type control (12-week-old C57BL/6 mice) shown in this study were consistent with those reported in previous studies by us [6] and another group [17].

## Evaluation of fibrosis

Among several quantitative methods that are available to determine interstitial fibrotic regions [15, 18, 19], we employed Masson-trichrome staining using the Accustatin Trichrome Stain Kit (#HT15-1KT; Sigma) in accordance with the manufacturer's protocol, as described previously [15, 20]. Interstitial fibrotic regions were quantified using image analysis software (Image J 1.45) to evaluate the percentage of blue area in the Masson-trichrome section [15].

## Evaluation of apoptosis

Apoptosis was determined by terminal deoxyribonucleotidyl transferase (TdT)-mediated biotin-16-deoxyuridine triphosphate (dUTP) nick-end labeling (TUNEL) staining using the Apoptosis *in situ* Detection Kit (#293–71501; FUJIFILM Wako, Osaka, Japan). TUNEL-positive nuclei per field of view were manually counted in six sections from each of the four groups (Control, PG-LPS, Cap and PG-LPS + Cap) over a microscopic field of 20 x, averaged and expressed as the ratio of TUNEL-positive nuclei (%) [15, 21]. Limiting the counting of total nuclei and TUNEL-positive nuclei to areas with true cross sections of myocytes made it possible to selectively count only those nuclei that were clearly located within myocytes.

## Western blotting

The cardiac muscle was excised from the mice (Fig 1A) and homogenized in a Polytron (Kinematica AG, Lucerne, Switzerland) in ice-cold RIPA buffer (Thermo Fisher Scientific, Waltham, MA, USA: 25 mM Tris-HCl (pH 7.6), 150 mM NaCl, 1% NP-40, 1% sodium deoxycholate, 0.1% SDS) without addition of inhibitors [22]. The homogenate was centrifuged at 13,000 x *g* for 10 min at 4˚C and the protein concentration in the supernatant was measured using a DC protein assay kit (Bio-Rad, Hercules, CA, USA). Equal amounts of protein (5 μg)

were subjected to 12.5% SDS-polyacrylamide gel electrophoresis and blotted onto 0.2 mm PVDF membrane (Millipore, Billerica, MA, USA).

Western blotting was conducted with commercial available antibodies [15, 21, 23]. The primary antibodies against α-smooth muscle actin (α-SMA) (1:1000, #19245), CaMKII (1:1000, #3362), phospho-CaMKII (1:1000, Thr-286, #3361), phospho-protein kinase C (PKC) δ (1:1000, Tyr-311, #2055) and PKCδ (1:1000, #2058) were purchased from Cell Signaling Technology (Boston, MA, USA), the primary antibodies against glyceraldehyde 3-phosphate dehydrogenase (GAPDH) (1:200, sc-32233) wss purchased from Santa Cruz Biotechnology (Santa Cruz, CA, USA), the primary antibodies against phospho-phospholamban (PLB) (1:5000, Thr-17, #A010-13) and PLB (1:5000, #A010-14), were purchased from Badrilla (Leeds,UK), and the primary antibodies against nicotinamide adenine dinucleotide phosphate oxidase-4 (NOX4), 1:1000, #ab133303) and xanthine oxidase (XO) (1:1000, #ab109235) were purchased from Abcam (Cambridge, UK). Horseradish peroxide-conjugated anti-rabbit (1:5000, #NA934) or anti-mouse IgG (1:5000, #NA931) purchased from GB Healthcare (Piscataway, NJ, USA) was used as a secondary antibody. The primary and secondary antibodies were diluted in Tris-buffered saline (pH 7.6) with 0.1% Tween 20 and 5% bovine serum albumin or immunoreaction enhancer solution (Can Get Signal; TOYOBO, #NKB-101, Osaka, Japan). The blots were visualized with enhanced chemiluminescence solution (ECL: Prime Western Blotting Detection Reagent, GE Healthcare) and scanned with a densitometer (ASL-600, GE Healthcare). Note that the blots of phospho-PKCδ (Tyr-311) and phospho-CaMKII (Thr-286) had many additional bands. Therefore, to identify appropriate bands for quantification, chronic infusion of Ang II (#015–27911; FUJIFILM Wako) dissolved in saline was performed for 7 days at a dose of 1,500 ng/kg/min via an osmotic mini-pump (Model 2001; AlZET, Cupertino, CA, USA) and the heart was isolated after 7 days [24, 25]. Control mice received infusion of a compatible volume of saline (**S1 Fig** of **S1 Data**). Bands whose density was increased in the treated mice were selected for quantification.

## Statistical analysis

Data are prepared as means ± standard deviation (SD). Comparison of data was performed using one-way ANOVA followed by Tukey's *post hoc* test. Differences were considered significant when $P < 0.05$.

We calculated the required total sample size of animals (α risk = 0.05, power (1-β) = 0.8) not only *a priori* (effect size ($f$) = 0.4) but also *a posteriori* with the effects size ($f$) derived *post hoc* [26] by means of G* Power version 3.1 (program, concept and design by Franz, Universitat Kiel, Germany; freely available Windows application software) [27] (**S2 Data**).

## Results

### Daily consumption of food and water

The consumed amounts of both food and water were similar among the four groups (food: PG-LPS [$n = 7$]: 10.7 ± 0.5, Cap [$n = 6$]: 9.0 ± 0.2, PG-LPS + Cap [$n = 7$]: 9.7 ± 0.7, all not significantly different [NS; $P > 0.05$] vs. Control [$n = 6$]; 9.8 ± 0.4 g/day; water: PG-LPS [$n = 7$]: 4.5 ± 0.3, Cap [$n = 6$]: 4.0 ± 0.1, PG-LPS + Cap [$n = 6$]: 3.9 ± 0.2, all NS [$P > 0.05$] vs. Control [$n = 6$]; 4.4 ± 0.1 mL/day) (**Fig 1B**).

### Effects of PG-LPS on body weight

The Control, PG-LPS, Cap, PG-LPS + Cap groups all showed similar BW at 1 week after the PG-LPS infusion (PG-LPS [$n = 7$]: 23 ± 0.5, Cap [$n = 6$]: 24 ± 0.4, PG-LPS + Cap [$n = 7$]:

24 ± 0.3 g, all NS [$P > 0.05$] vs. Control [$n = 6$; 24 ± 0.4 g]) (**Fig 1C**). These data, together with the data shown in **Fig 1B**, suggest that chronic PG-LPS treatment with or without Cap under the experimental conditions used in this study had no significant effect on growth, food consumption or water consumption, although we cannot rule out the possibility that the apparent lack of difference was due to the relatively small number of animals used.

## Effects of PG-LPS on cardiac function

We conducted echocardiography (**Table 1**) to evaluate cardiac function in terms of left ventricular ejection fraction (EF) and fractional shortening (%FS). Both parameters were significantly decreased in the PG-LPS group compared to the control (EF: Control [$n = 6$] vs. PG-LPS [$n = 7$]: 66 ± 1.8 vs. 59 ± 2.5%, $P < 0.01$; %FS: Control [n = 6] vs. PG-LPS [$n = 7$]: 31 ± 1.2 vs. 27 ± 1.5%, $P < 0.01$). Importantly, PG-LPS-mediated cardiac dysfunction was attenuated by pharmacological inhibition of RAS with Cap at 1 week (EF: PG-LPS [$n = 7$] vs. PG-LPS + Cap [$n = 6$]: 59 ± 2.5 vs. 63 ± 1.1%, $P < 0.05$; %FS: PG-LPS [$n = 7$] vs. PG-LPS + Cap [$n = 6$]: 27 ± 1.5 vs. 29 ± 0.7%, $P < 0.05$).

**Table 1. Cardiac function assessed by echocardiography at 1 week after PG-LPS.**

|  | Control | BO | Cap | BO + Cap |
|---|---|---|---|---|
| n | 6 | 7 | 6 | 6 |
| EF | 66 ± 1.8 | 59 ± 2.5** | 64 ± 2.7## | 63 ± 1.1# |
| EDV | 0.22 ± 0.02 | 0.21 ± 0.03 | 0.21 ± 0.02 | 0.20 ± 0.01 |
| ESV | 0.07 ± 0.006 | 0.08 ± 0.014 | 0.07 ± 0.004 | 0.07 ± 0.013 |
| %FS | 31 ± 1.2 | 27 ± 1.5** | 30 ± 1.8## | 29 ± 0.7# |
| LVIDd | 4.45 ± 0.13 | 4.36 ± 0.20 | 4.36 ± 0.11 | 4.35 ± 0.06 |
| LVIDs | 3.1 ± 0.08 | 3.2 ± 0.19 | 3.0 ± 0.06 | 3.1 ± 0.02 |
| HR | 443 ± 63.7 | 431 ± 39.2 | 428 ± 54.7 | 407 ± 54.1 |
| SV | 0.14 ± 0.01 | 0.12 ± 0.01* | 0.13 ± 0.01 | 0.13 ± 0.007 |
| CO | 58 ± 8.1 | 47 ± 5.0 | 52 ± 8.6 | 49 ± 6.3 |
| IVSTd | 0.5 ± 0.04 | 0.5 ± 0.04 | 0.47 ± 0.03 | 0.5 ± 0.04 |
| LVSTs | 0.9 ± 0.10 | 0.83 ± 0.04 | 0.86 ± 0.06 | 0.88 ± 0.06 |
| LVPWTd | 0.49 ± 0.04 | 0.47 ± 0.03 | 0.5 ± 0.05 | 0.47 ± 0.03 |
| LVPWTs | 0.9 ± 0.10 | 0.79 ± 0.03* | 0.88 ± 0.06 | 0.85 ± 0.04 |

EF (%): left ventricular ejection fraction.

EDV (mL): left ventricular end-diastolic volume.

ESV (mL): left ventricular end-systolic volume.

%FS: % fractional shortening.

LVIDd (mm): left ventricular internal dimension at end-diastole.

LVIDs (mm): left ventricular internal dimension at end-diastole.

HR (bpm): heart rate.

SV (mL): stroke volume.

CO (mL/min): cardiac output.

IVSTd (mm): interventricular septum thickness at end-diastole.

LVSTs (mm): interventricular septum thickness at end-systole.

LVPWTd (mm): left ventricular posterior wall thickness at end-diastole.

LVPWTs (mm): left ventricular posterior wall thickness at end-diastole.

**P < 0.01 vs. Control

*P < 0.05 vs. Control

## P < 0.01 vs. LPS

# P < 0.05 vs. LPS

These data suggest that PG-LPS treatment might mediate cardiac dysfunction through activation of the RAS.

## Effects of PG-LPS on serum Ang II levels

The renin-angiotensin system is thought to be involved in inflammatory processes such as periodontitis. However, its precise role is still unclear. Therefore, we next examined serum levels of Ang II in the four groups by mean of ELISA. Serum Ang II levels were significantly increased in the PG-LPS group compared to the control (Control [$n = 4$] vs. PG-LPS [$n = 4$]: 88 ± 12 vs. 759 ± 624 pg/ml, $P < 0.05$ by one-way ANOVA followed by the Tukey-Kramer *post hoc* test), and the increase was significantly suppressed by Cap (PG-LPS [$n = 4$] vs. PG-LPS + Cap [$n = 5$]: 759 ± 624 vs. 62 ± 67 pg/ml, $P < 0.05$ by one-way ANOVA followed by the Tukey-Kramer *post hoc* test (**Fig 2A**).

These results suggest that activation of RAS might be mediated by PG-LPS treatment.

## Effects of PG-LPS on heart size, lung weight and liver weight

We examined the effect of PG-LPS with/without Cap on heart size in terms of cardiac muscle mass per tibial length ratio (**Fig 2B**), as well as the effects on wet lung and liver weight per tibial

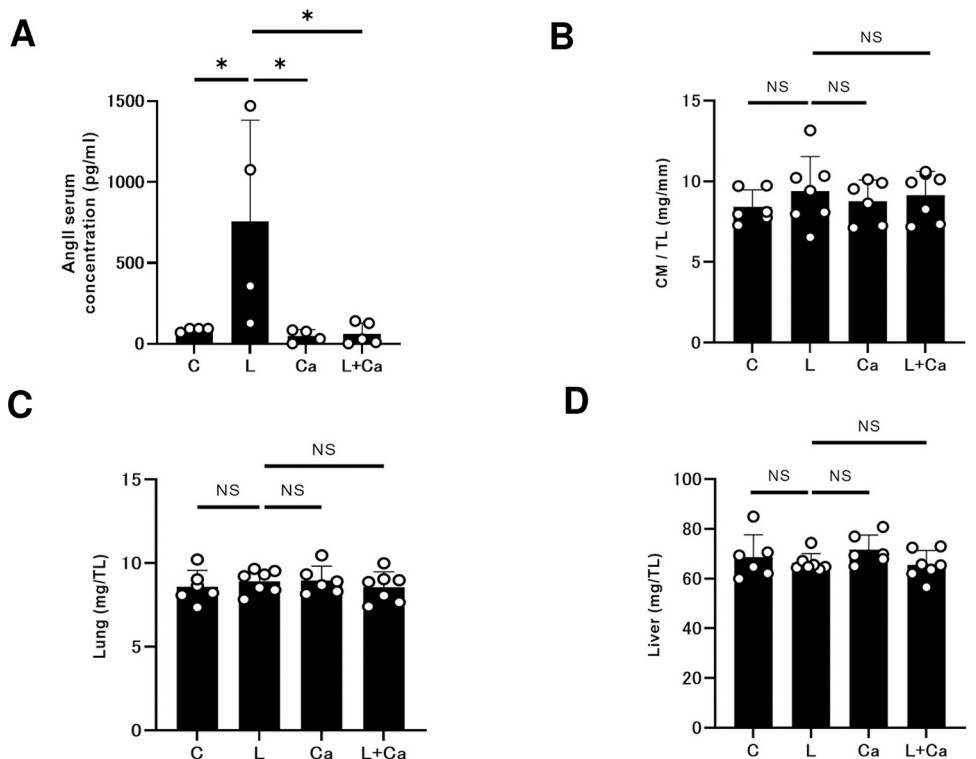

**Fig 2. Effects of chronic PG-LPS infusion on Ang II and the weights of cardiac muscle, lung and liver. (A)** Serum Ang II levels were significantly increased in the PG-LPS group (L), and this increase was significantly attenuated by Cap (L + Ca). *$P < 0.05$ vs. Control (C), *$P < 0.05$ vs. PG-LPS group (L) or *$P < 0.05$ vs. PG-LPS + Cap group (L + Ca) by one-way ANOVA followed by the Tukey-Kramer *post hoc* test. Data shows means ± SD and open circles show individual data. **(B-D)** Cardiac muscle (CM) weight per tibia length (TL) ratio **(B)**, lung weight per TL ratio **(C)**, and liver weight per TL ratio **(D)** were all similar among the Control (C), PG-LPS (L), Cap (Ca) and LPS + Cap (L + Ca) groups. NS, not significantly different from the Control ($P > 0.05$) by one-way ANOVA followed by the Tukey-Kramer *post hoc* test.

length ratio (**Fig 2C and 2D**). Similar results were obtained among the four groups (Control: *n* = 6; PG-LPS: *n* = 7; Cap: *n* = 6; PG-LPS + Cap: *n* = 7).

These data suggest that PG-LPS did not induce cardiac hypertrophy, lung edema or liver congestion during the experimental period.

## Effects of PG-LPS on cardiac fibrosis

We examined the effects of PG-LPS with/without Cap on fibrosis in cardiac muscle by means of Masson-trichrome staining (**Fig 3A**). PG-LPS treatment significantly increased the area of fibrosis in cardiac muscle (Control [*n* = 6] vs. PG-LPS [*n* = 7]: $0.7 \pm 0.1$ vs. $2.0 \pm 0.2\%$, $P < 0.01$ by one-way ANOVA followed by the Tukey-Kramer *post hoc* test), in accordance with our previous findings [3, 6] (**Fig 3B**). Cap alone did not alter the area of fibrosis, but it blocked the BO-induced increase of fibrosis (PG-LPS [*n* = 7] vs. PG-LPS + Cap [*n* = 7]:

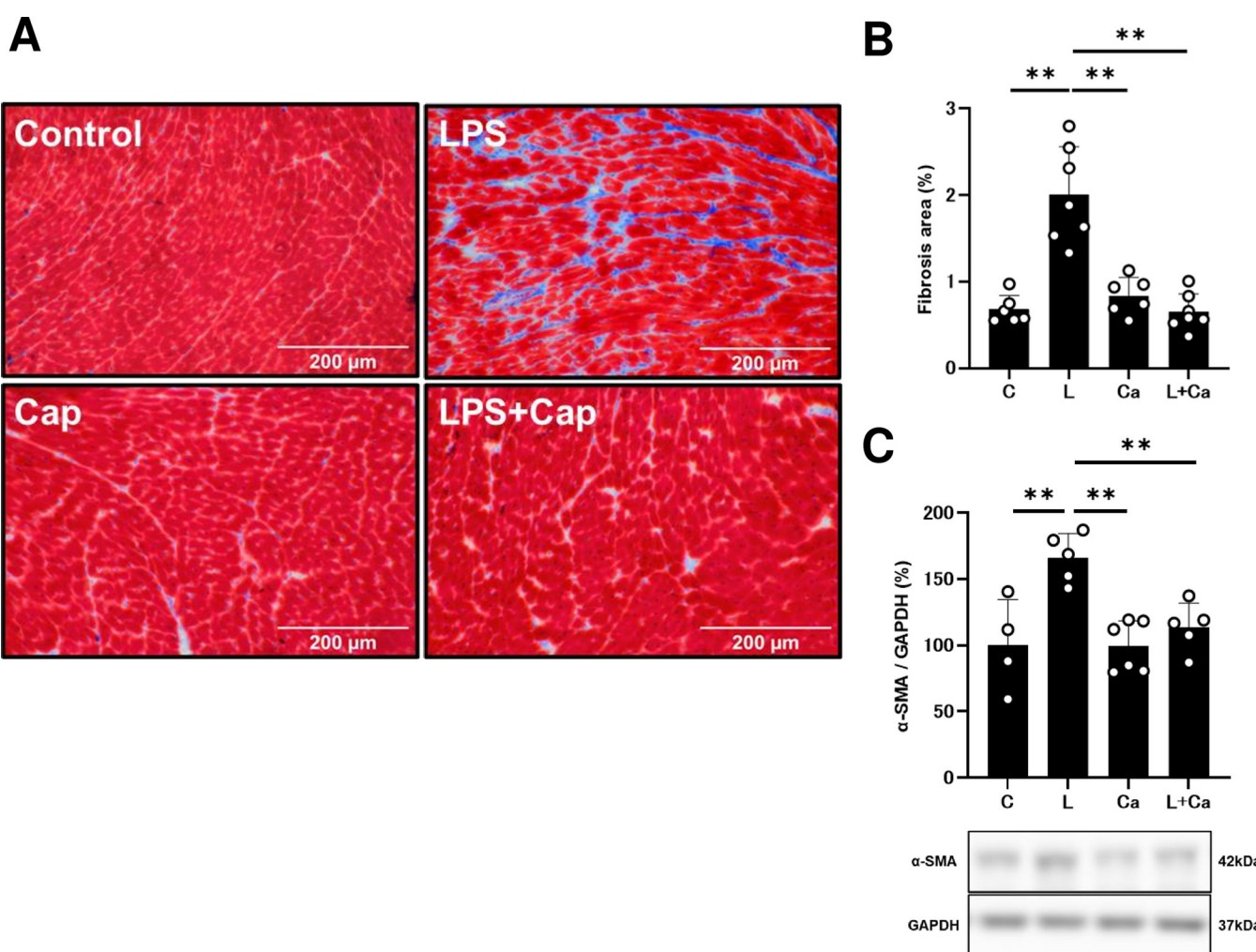

**Fig 3. Effects of chronic PG-LPS infusion on fibrosis in cardiac muscle. (A)** Representative images of Masson-trichrome-stained sections of cardiac muscle from the Control (***upper left***), PG-LPS (LPS) (***upper right***), Cap (***lower left***) and PG-LPS + Cap (LPS + Cap) (***lower right***) groups. Scale bars: 200 μm. **(B)** The area of fibrosis was significantly increased in the PG-LPS group (L) ($P < 0.01$ vs. Control), and this increase was significantly attenuated by Cap (L + Ca). **$P < 0.01$ vs. Control (C), **$P < 0.01$ vs. PG-LPS group (L) or **$P < 0.01$ vs. PG-LPS + Cap group (L + Ca) by one-way ANOVA followed by the Tukey-Kramer *post hoc* test. **(C)** Expression of α-SMA was significantly increased in cardiac muscle of PG-LPS-treated mice (L), and the increase was significantly attenuated by Cap (L + Cap). **$P < 0.01$ vs. Control (C), **$P < 0.01$ vs. PG-LPS group (L) or **$P < 0.01$ vs. PG-LPS + Cap group (L + Ca) by one-way ANOVA followed by the Tukey-Kramer *post hoc* test. Full-size images of the immunoblots are presented in **S2 Fig** of **S1 Data**. Data shows means ± SD and open circles show individual data.

2.0 ± 0.2 vs. 0.6 ± 0.1%, $P < 0.01$ by one-way ANOVA followed by the Tukey-Kramer *post hoc* test) (**Fig 3B**).

## Effects of PG-LPS on α-SMA expression

We also evaluated cardiac fibrosis by measuring the level of α-SMA expression at 1 week after the start of PG-LPS, because this parameter is closely associated with cardiac fibrosis [3, 6, 7, 28]. Expression of α-SMA was significantly increased in cardiac muscle of PG-LPS-treated mice (Control [$n = 4$] vs. PG-LPS [$n = 5$]: 100 ± 17.3 vs. 166 ± 8.2%, $P < 0.01$ by one-way ANOVA followed by the Tukey-Kramer *post hoc* test), and the increase was significantly suppressed by Cap (PG-LPS [$n = 5$] vs. PG-LPS + Cap [$n = 5$]: 166 ± 8.2 vs. 114 ± 8.2%, $P < 0.01$ by one-way ANOVA followed by the Tukey-Kramer *post hoc* test) (**Fig 3C**).

These data support the idea that cardiac fibrosis induced by PG-LPS treatment might be mediated, at least in part, through activation of the RAS.

## Effects of PG-LPS on cardiac apoptosis

We next evaluated apoptosis of cardiac myocytes in PG-LPS-treated mice with/without Cap treatment by means of TUNEL staining (**Fig 4A**). PG-LPS treatment significantly increased cardiac myocyte apoptosis (Control [$n = 4$] vs. PG-LPS [$n = 4$]: 1.1 ± 0.1 vs. 6.2 ± 0.8%, $P < 0.01$ by one-way ANOVA followed by the Tukey-Kramer *post hoc* test). Cap alone ($n = 4$)

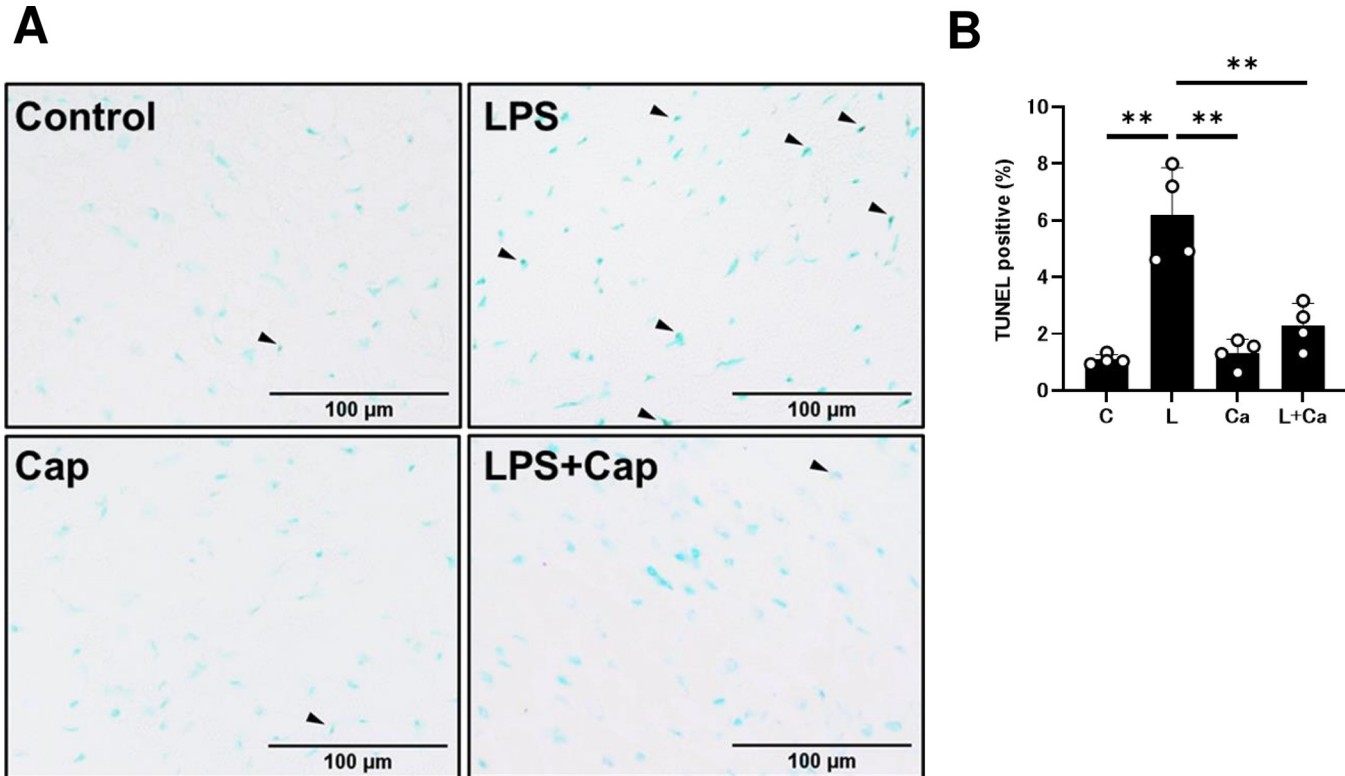

**Fig 4. Effects of chronic PG-LPS infusion on cardiac myocyte apoptosis. (A)** Representative images of TUNEL-stained sections of cardiac muscle from the Control (*upper left*), PG-LPS (LPS) (*upper right*), Cap (*lower left*) and PG-LPS + Cap (LPS + Cap) (*lower right*) groups. Scale bars: 100 μm. Black arrows: TUNEL-positive myocytes. **(B)** The number of TUNEL-positive myocytes was significantly increased in the PG-LPS group (L) ($P < 0.01$ vs. Control), and this increase was significantly attenuated by Cap (L + Ca). **P < 0.01 vs. Control (C), **P < 0.01 vs. PG-LPS group (L) or **P < 0.01 vs. PG-LPS + Cap group (L + Ca) by one-way ANOVA followed by the Tukey-Kramer *post hoc* test. Data shows means ± SD and open circles show individual data.

had no effect on the number of TUNEL-positive cardiac myocytes, but it blocked the BO-induced increase of TUNEL-positive cardiac myocytes (PG-LPS [$n = 4$] vs. PG-LPS + Cap [$n = 4$]: 6.2 ± 0.8 vs. 2.3 ± 0.4%, $P < 0.01$ by one-way ANOVA followed by the Tukey-Kramer *post hoc* test) (**Fig 4B**).

These data suggest that the induction of cardiac myocyte apoptosis by PG-LPS treatment might be mediated, at least in part, through activation of the RAS.

## Effects of PG-LPS on PKCδ phosphorylation

Phosphorylation of PKCδ among the PKC isoforms expressed in the heart is important for the development of angiotensin II type 1 receptor (AT1R)-induced cardiac remodeling [29]. We first confirmed the identity of the band phosphorylated by Ang II, because phospho-PKCδ (Tyr-311) shows many additional bands (**S1A Fig** of **S1 Data**).

We therefore examined the phosphorylation status of tyrosine 311 of PKCδ in the heart in the four groups [29, 30]. Phosphorylation of tyrosine 311 was significantly increased in the PG-LPS-treated mice (Control [$n = 5$] vs. PG-LPS [$n = 5$]: 100 ± 5.0 vs. 148 ± 12.8%, $P < 0.05$ vs. Control), and the increase was suppressed by Cap (PG-LPS [$n = 5$] vs. PG-LPS + Cap [$n = 5$]; 148 ± 12.8 vs. 74 ± 6.4%, $P < 0.01$ vs. PG-LPS) (**Fig 5A**).

These data suggest that PG-LPS-induced cardiac fibrosis and myocyte apoptosis might be mediated by AT1-mediated phosphorylation of PKCδ at tyrosine 311, leading to cardiac dysfunction.

## Effects of PG-LPS on NOX4 and XO expression

Stimulation of AT1R results in cardiac reactive oxygen species (ROS) generation through a number of pathways, including NOX and XO, and may be involved in myocardial fibrosis and myocyte apoptosis [31].

Two NOX isoforms, NOX2 and NOX4, are expressed in the heart, and their activity is regulated by their expression level [32]. AT1-AR-induced cardiac fibrosis and remodeling might be caused by ROS production via AT1/NOX4 interaction [31]

We therefore examined NOX4 protein expression in the heart among the four groups. NOX4 expression was significantly increased in the heart of the PG-LPS-treated group (Control [$n = 5$] vs. PG-LPS [$n = 6$]: 100 ± 1.9 vs. 130 ± 11.7%, $P < 0.05$ vs. Control), and the increase was suppressed by Cap (PG-LPS [$n = 6$] vs. PG-LPS + Cap [$n = 6$]; 130 ± 11.7 vs. 94 ± 5.0%, $P < 0.01$ vs. PG-LPS) (**Fig 5B**).

We next examined XO expression in the heart in the four groups because its activity is also regulated by its expression level [33, 34]. XO expression was significantly increased in the PG-LPS-treated group (Control [$n = 6$] vs. PG-LPS [$n = 7$]: 100 ± 17.9 vs. 415 ± 55%, $P < 0.01$ vs. Control), and the increase was suppressed by Cap (PG-LPS [$n = 7$] vs. PG-LPS + Cap [$n = 7$]; 415 ± 55 vs. 173 ± 32%, $P < 0.01$ vs. PG-LPS) (**Fig 5C**).

These data suggest that PG-LPS-induced ROS generation might contribute, at least in part, to the upregulation of ROS-producing enzymes: NOX4 and XO.

## Effects of PG-LPS on CaMKII phosphorylation

CaMKII is activated via phosphorylation in the presence of ROS and contributes to the development of cardiac remodeling and dysfunction [6–8, 35].

We thus examined the amounts of phospho-CaMKII (Thr-286) in the heart in the four groups. We first confirmed the identity of the bands phosphorylated by Ang II, because phospho-CaMKII (Tyr-286) shows many additional bands (**S1B Fig** of **S1 Data**).

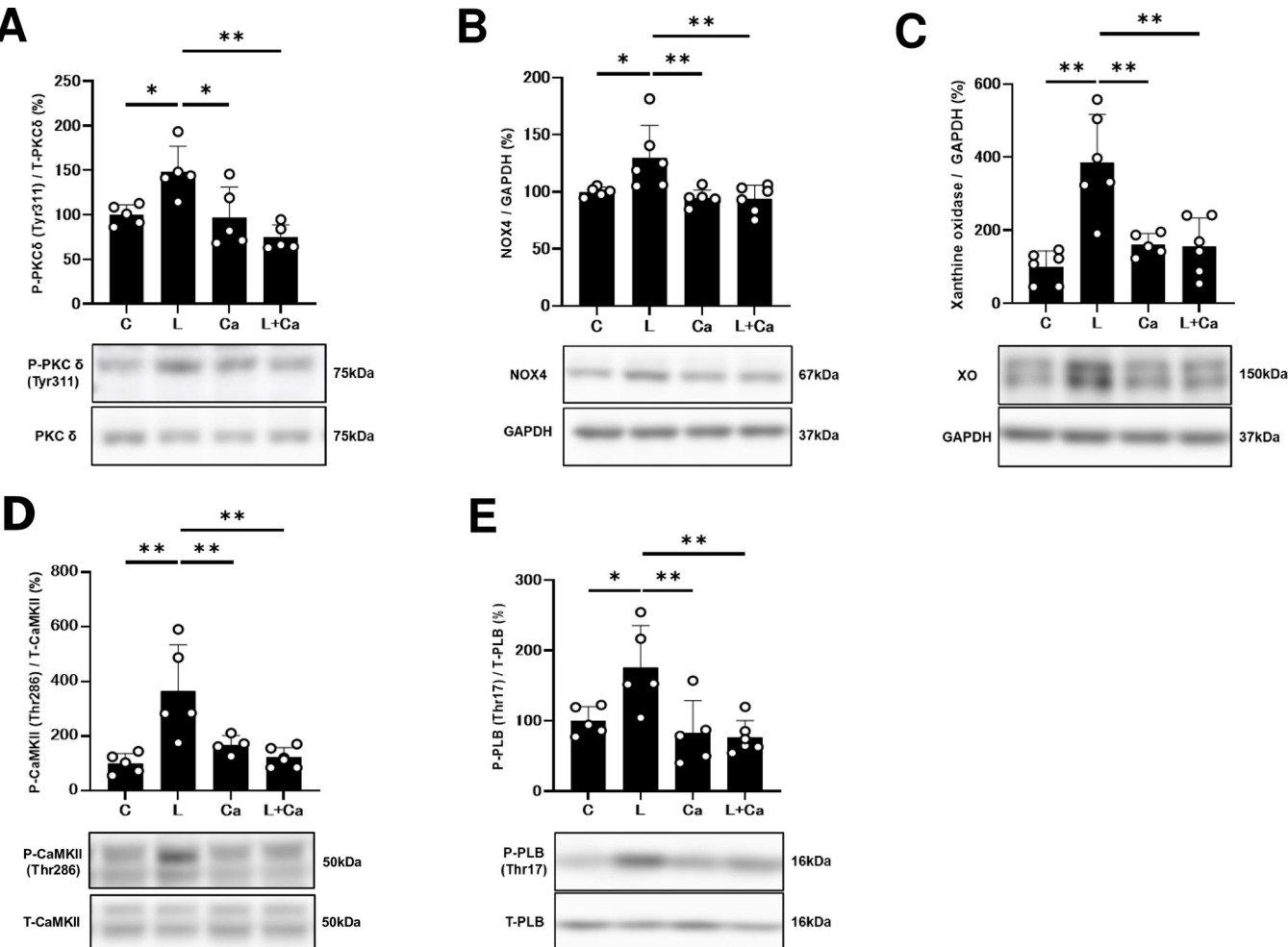

**Fig 5.** **(A)** PKCδ phosphorylation (Tyr-311) was significantly increased in the PG-LPS group (L), and this increase was significantly attenuated by Cap (L + Ca). *$P < 0.05$ vs. Control (C), *$P < 0.05$ vs. PG-LPS group (L) or **$P < 0.01$ vs. PG-LPS + Cap group (L + Ca) by one-way ANOVA followed by the Tukey-Kramer *post hoc* test. Full-size images of the immunoblots are presented in **S3 Fig** of **S1 data**. **(B)** NOX4 expression was significantly increased in the PG-LPS group (L), and this increase was significantly attenuated by Cap (L + Ca). **$P < 0.05$ vs. Control (C), **$P < 0.01$ vs. PG-LPS group (L) or **$P < 0.01$ vs. PG-LPS + Cap group (L + Ca) by one-way ANOVA followed by the Tukey-Kramer *post hoc* test. Full-size images of the immunoblots are presented in **S4 Fig** of **S1 Data**. **(C)** XO expression was significantly increased in the PG-LPS group (L), and this increase was significantly attenuated by Cap (L + Ca). **$P < 0.01$ vs. Control (C), **$P < 0.01$ vs. PG-LPS group (L) or **$P < 0.01$ vs. PG-LPS + Cap group (L + Ca) by one-way ANOVA followed by the Tukey-Kramer *post hoc* test. Full-size images of the immunoblots are presented in **S5 Fig** of **S1 Data**. **(D)** CaMKII phosphorylation (Thr-286) was significantly increased in the PG-LPS group (L), and this increase was significantly attenuated by Cap (L + Ca). **$P < 0.01$ vs. Control (C), **$P < 0.01$ vs. PG-LPS group (L) or **$P < 0.01$ vs. PG-LPS + Cap group (L + Ca) by one-way ANOVA followed by the Tukey-Kramer *post hoc* test. Full-size images of the immunoblots are presented in **S6 Fig** of **S1 data**. **(E)** PLB phosphorylation (Thr-17) was significantly increased in the PG-LPS group (L), and this increase was significantly attenuated by Cap (L + Ca). *$P < 0.05$ vs. Control (C), **$P < 0.01$ vs. PG-LPS group (L) or **$P < 0.01$ vs. PG-LPS + Cap group (L + Ca) by one-way ANOVA followed by the Tukey-Kramer *post hoc* test. Full-size images of the immunoblots are presented in **S7 Fig** of **S1 Data**. Data shows means ± SD and open circles show individual data.

We found significant increase in the PG-LPS-treated group (Control [$n = 5$] vs. PG-LPS [$n = 5$]: 100 ± 16.2 vs. 364 ± 75.9%, $P < 0.01$ vs. Control). The increase was suppressed by Cap (PG-LPS [$n = 5$] vs. PG-LPS + Cap [$n = 6$]; 364 ± 75.9 vs. 122 ± 14.6%, $P < 0.01$ vs. PG-LPS) (**Fig 5D**).

These data suggest that CaMKII signaling might be activated by PG-LPS infusion through NOX4- and XO-mediated generation of ROS, at least in part, via activation of the RAS.

### Effects of PG-LPS on PLB phosphorylation

Since phosphorylation of most $Ca^{2+}$-handling proteins is altered in many models of experimental heart failure and might lead to increased $Ca^{2+}$ leakage, we next examined the effects of PG-LPS treatment on PLB phosphorylation at Thr-17, which is known to be mediated by CaMKII [15].

Phospho-PLB (Thr-17) was significantly increased in the heart of the PG-LPS-treated group (Control [$n = 5$] vs. PG-LPS [$n = 5$]: 100 ± 9.0 vs. 176 ± 26.5%, $P < 0.01$ vs. Control) (**Fig 5E**). This increase was significantly attenuated by Cap (PG-LPS [$n = 5$] vs. PG-LPS + Cap [$n = 6$]; 176 ± 26.5 vs. 77 ± 9.6%, $P < 0.01$ vs. PG-LPS)

Together with previous findings [36, 37], these data suggest that PG-LPS might increase PLB phosphorylation at least in part through activation of the RAS, leading to ROS-mediated elevation of diastolic sarcoplasmic reticulum $Ca^{2+}$ leakage in cardiac myocytes.

## Discussion

Oral health is important for maintaining general health among the elderly. A number of epidemiological studies have found links between poor oral health and a range of medical conditions including cardiovascular disease [38], type 2 diabetes [39], adverse pregnancy outcome [40], osteoporosis [41], aspiration pneumonia [42] and rheumatoid arthritis [43]. However, the longitudinal association between poor oral health and general health has not been reported. Instead, the recent focus has been on identifying possible mechanisms that underlie these associations and on examining whether treating oral disease leads to an improvement in markers of systemic disease. Cardiovascular disease, obesity and diabetes in particular, are significant public health problems worldwide and governments are well aware that, unless action is taken, the cost of treating patients with these conditions is likely to become unmanageable. Poor oral health is also a public health problem, with gingivitis and chronic periodontitis being among the most common human infections.

Our findings here indicate that cardiac function was significantly impaired in mice treated with PG-LPS at a dose consistent with circulating levels in periodontitis patients, and myocyte apoptosis and fibrosis were significantly increased. Importantly, these changes were blunted by pharmacological inhibition of the RAS with the ACE inhibitor Cap. We then investigated the mechanisms of these changes.

We have recently demonstrated that persistent subclinical exposure to PG-LPS induces cardiac dysfunction, myocyte apoptosis and fibrosis in cardiac muscle via activation of toll-like receptor 4 (TLR4) signaling in mice [6]. Activation of the innate immune system in the heart by TLR4 has diverse effects, which may be cardioprotective in the short term, though sustained activation may be maladaptive [44]. However, the mechanisms remain incompletely understood, even though subclinical endotoxemia was previously reported to induce myocardial cell damage and heart failure [45]. TLR4 activation was recently demonstrated to be involved in activation of the RAS system induced by uric acid in adipose tissue, causing hypertension and higher expression levels of inflammatory cytokines [46]. In addition, the interplay of TLR4 signaling and the RAS might contribute to the pathogenesis of diabetic nephropathy [47]. We thus anticipated that cardiac dysfunction induced by persistent subclinical exposure to PG-LPS might be caused via activation of the RAS system, and our present findings support this hypothesis.

TLR4 signaling was recently demonstrated to participate in sympathetic hyperactivity in the paraventricular nucleus [48]. In addition, we recently reported that the cAMP/protein kinase A and cAMP/CaMKII signaling pathways were activated in the heart of PG-LPS-treated mice, as employed in the present study [3]. Activation of both the SNS and RAS is associated

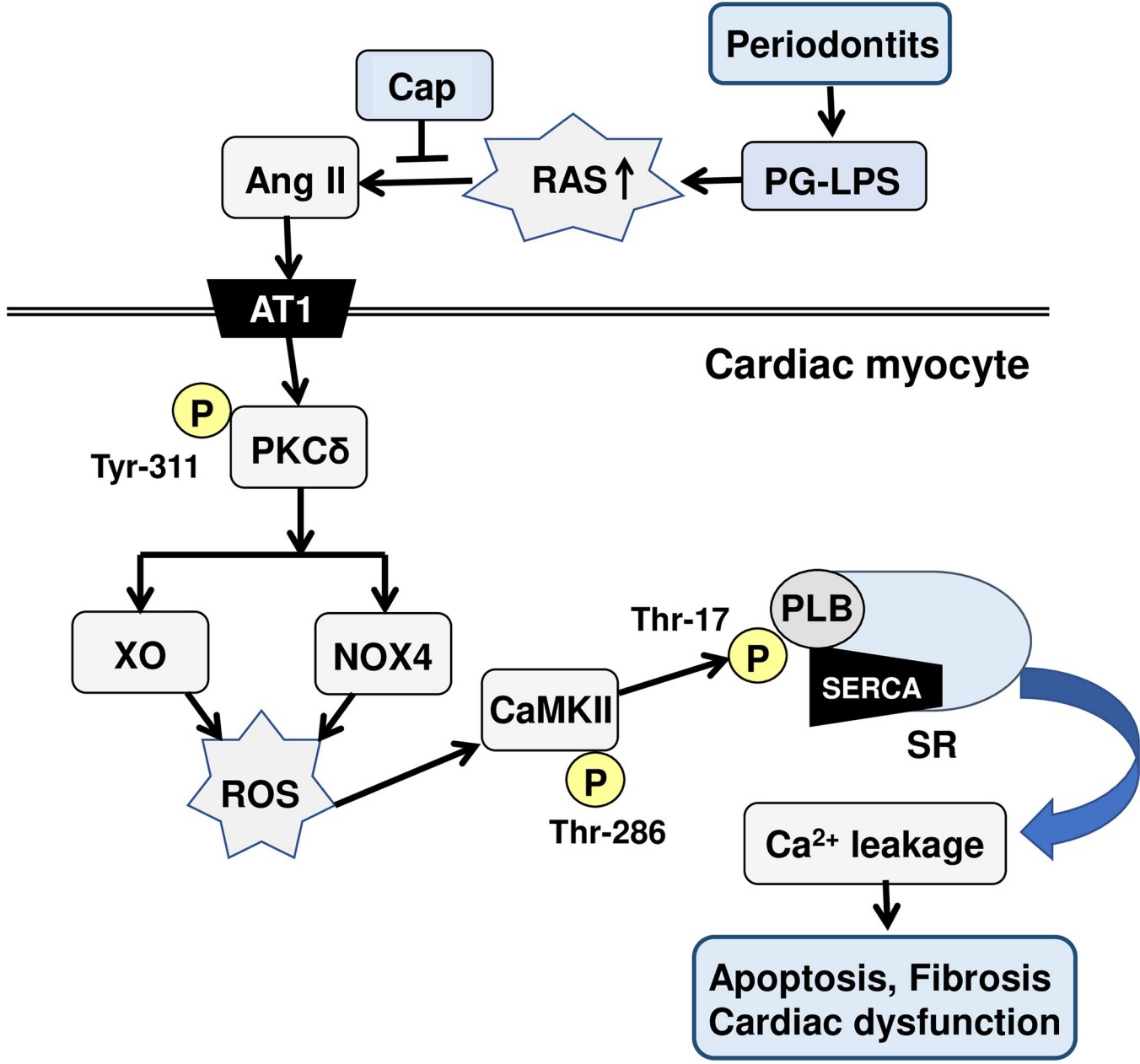

**Fig 6. This scheme illustrates the proposed role of RAS in the heart of PG-LPS-treated mice.** PG-LPS-treatment induces phosphorylation on PKCδ (Tyr-311), leading to CaMKII activation with the increased ROS production via activation of NOX4/XO. PG-LPS-treatment was previously reported to induce sympathetic nerve activity by our group, leading to the phosphorylation of PLB (Thr-17). These changes might cause $Ca^{2+}$ leakage, leading to fibrosis, myocyte apoptosis, oxidative stress and cardiac dysfunction.

with cardiovascular diseases, including heart failure, though the interactions of the many factors involved in these two systems remain unclear [49]. Adenylyl cyclase (AC) is the target enzyme of β-adrenergic receptor (β-AR) signaling stimulation. At least 9 isoforms are known and the type 5 isoform (AC5) is a major AC isoform not only in cardiac myocytes [50], but also in juxtaglomerular cells (JG cells) [51]. Renin is a key component of the RAS system, and cAMP generated by AC5 stimulates renin gene transcription in JG cells. We hypothesized that inhibition of RAS with Cap would have a protective effect against PG-LPS-induced cardiac

dysfunction. In this study, we obtained the first evidence that pharmacological inhibition with Cap can protect the heart from PG-LPS-induced cardiac fibrosis, myocyte apoptosis, and cardiac dysfunction.

PKC consists of at least 11 isoforms and is a major target of Ang II [52, 53]. PKC isoforms are expressed differentially in various organs and the major PKC isoforms expressed in murine heart are PKCδ and PKCε [54], though their roles are temporally distinct and opposite. PKCε plays a key role in cardioprotection as a result of intracellular translocation from cytosol to the membrane fraction [53]. On the other hand, PKCδ exists in the cytoplasm and is activated by tyrosine phosphorylation [29, 55]. Several tyrosine phosphorylation sites exist in the catalytic domain (Tyr-512 and Tyr-523), regulatory domain (Tyr-52, Tyr-155, Tyr-187), and hinge region (Tyr-311 and Tyr-332) [30]. However, PKCδ phosphorylated at Tyr-311 shows enhanced catalytic activity in vitro [56] and plays a role of in promoting cardiac fibrosis, myocyte apoptosis and heart failure [57]. We thus examined the phosphorylation level of PKC on Tyr-311 and found that it was significantly increased in the PG-LPS-treated group and Cap significantly suppressed its phosphorylation, indicating that PKC phosphorylation at Tyr-311 might be important for PG-LPS-mediated cardiac remodeling and cardiac dysfunction.

Increased RAS function was demonstrated to induce marked activation of PKCδ in podocytes, resulting in increased ROS production [58]. In this study, we confirmed that the expression levels of ROS-producing enzymes (NOX4 and XO) were significantly increased concomitantly with the phosphorylation of PLB at a CaMKII-dependent phosphorylation site. These findings suggest that PG-LPS at a dose consistent with circulating levels in periodontal patients increases the expression of ROS-producing enzymes, which in turn might augment PG-LPS-induced cardiac dysfunction in association with increased PKCδ phosphorylation, leading to ROS production and abnormal $Ca^{2+}$ release from the sarcoplasmic reticulum (**Fig 6**).

The involvement of the RAS in myocardial fibrosis is evident in several pathological conditions such as hypertensive heart disease [59], congestive heart failure [60] and myocardial infarction [61]. It has been suggested that periodontitis may be a contributory risk factor for cardiovascular disease [62]. Although many studies have found an association between periodontitis and cardiovascular disease, the relationship is still controversial and is thought to require more investigation [63, 64]. To our knowledge, this study is the first to establish an association between periodontis and myocardial fibrosis and to afford evidence that RAS might be a key mediator of this association.

Overall, our results, together with the previous findings, suggest that the ACE inhibitor Cap inhibits the onset of cardiovascular disease in periodontal patients, and therefore might be helpful for maintaining general health and improving longevity.

## Supporting information

**S1 Data. Representative full-length immunoblots shown in the main article.**
(PDF)

**S2 Data. Tables of effect sizes, sample sizes and statistical power.**
(PDF)

## Author Contributions

**Conceptualization:** Kenichi Kiyomoto, Ichiro Matsuo, Kenji Suita, Yoshiki Ohnuki, Satoshi Okumura.

**Formal analysis:** Kenichi Kiyomoto, Ichiro Matsuo, Yasuharu Amitani, Satoshi Okumura.

**Funding acquisition:** Ichiro Matsuo, Kenji Suita, Yoshiki Ohnuki, Aiko Ito, Megumi Nar-iyama, Satoshi Okumura.

**Investigation:** Kenichi Kiyomoto, Ichiro Matsuo, Kenji Suita, Michinori Tsunoda, Akinaka Morii, Yoshio Hayakawa.

**Methodology:** Kenichi Kiyomoto, Ichiro Matsuo, Kenji Suita, Yoshiki Ohnuki, Misao Ishi-kawa, Yasumasa Mototani.

**Supervision:** Kazuhiro Gomi, Satoshi Okumura.

**Writing – original draft:** Kenichi Kiyomoto, Ichiro Matsuo, Satoshi Okumura.

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
