## [Decision Letter · Decision Letter 0]

27 Feb 2023

PONE-D-23-02675Oral angiotensin-converting enzyme inhibitor captopril protects the heart from Porphyromonas gingivalis LPS-induced cardiac dysfunction in micePLOS ONE

Dear Dr. Okumura,

Thank you for submitting your manuscript to PLOS ONE. After careful consideration, we feel that it has merit but does not fully meet PLOS ONE’s publication criteria as it currently stands. Therefore, we invite you to submit a revised version of the manuscript that addresses the points raised during the review process. In particular, you should supply the original Western blots to allow the evaluation of antibody specificity. ]Please submit your revised manuscript by Apr 13 2023 11:59PM. If you will need more time than this to complete your revisions, please reply to this message or contact the journal office at plosone@plos.org. Please include the following items when submitting your revised manuscript:A rebuttal letter that responds to each point raised by the academic editor and reviewer(s). You should upload this letter as a separate file labeled 'Response to Reviewers'.A marked-up copy of your manuscript that highlights changes made to the original version. You should upload this as a separate file labeled 'Revised Manuscript with Track Changes'.An unmarked version of your revised paper without tracked changes. You should upload this as a separate file labeled 'Manuscript'.

We look forward to receiving your revised manuscript.

Kind regards,

Michael Bader

Academic Editor

PLOS ONE

Reviewers' comments:

Reviewer's Responses to Questions

**Comments to the Author**

1. Is the manuscript technically sound, and do the data support the conclusions?

Reviewer #1: Partly

Reviewer #2: No

Reviewer #3: Yes

2. Has the statistical analysis been performed appropriately and rigorously? 

Reviewer #1: No

Reviewer #2: I Don't Know

Reviewer #3: Yes

3. Have the authors made all data underlying the findings in their manuscript fully available?

Reviewer #1: Yes

Reviewer #2: No

Reviewer #3: Yes

4. Is the manuscript presented in an intelligible fashion and written in standard English?

Reviewer #1: Yes

Reviewer #2: Yes

Reviewer #3: Yes

5. Review Comments to the Author

Reviewer #1: The present study aimed to evaluate the possible protective effects of oral Captopril in mice hearts infected with Porphyromonas gingivalis LPS (LPS-Pg), i.p. The relationship between poor oral health and cardiovascular complications, such as heart failure and fibrosis, is widespread in the literature. The important role of RAS components in cardiac functions and structures is also known, especially when it comes to Ang-II peptide linked to AT1 receptor.

It was the hypothesis of this study that there would be cardiovascular protection promoted by Captopril with regard to decrease in cardiac fibrosis caused by LPS-Pg, as well as an improvement in ejection fraction also in mice treated with LPS-Pg. However, the sample size calculation for carrying out this study was not presented, so there is no guarantee that the number of mice used reflects the reality of the results.

In addition, the figure 6 concludes that the increase in oxidative stress with consequent increase in apoptosis and cardiac fibrosis, as well as the left ventricular ejection fraction worsening, is linked to Ang-II and AT1 receptors interaction. Bearing this in mind, the Captopril use could decrease the Ang-II availability and contribute to the improvement of cardiac function in mice infected with LPS-Pg. On the other hand, there is no mention in this study other possibilities of Ang-II formation by other enzymatic pathways besides the Converse Angiotensin Enzyme (ACE) function. Elastase-2, for example, can cleave angiotensinogen directly into Ang-II, so an Ang-II dosage throughout the study is essential for figure 6 conclusion. Finally, it would be interesting and important to have access to complete Western Blot membranes photos of all the targets studied, because in some cases, such as the markings for BCL-2, AT-1 and XO, the bands shown are double or triple and this can configure low antibodies selectivity and efficiency. It is worth mentioning that the presented images blots are cropped.

Reviewer #2: The goal of this study was to investigate the effects of the ACE inhibitor captopril on Porphyromonas gingivalis lipopolysaccharide (PG-LPS) induced cardiac dysfunction in mice. The findings of this study are interesting and align with studies evaluating the effects of oral disease state in the heart however, the conclusion that RAS is involved is a jump that is not substantiated by the data. In addition, there is clear issues with the data presented specifically with the echo and immunoblotting.

1) Immunoblots are not convincing as they are highly zoomed in and blurry. In some cases, the images do not visually look different despite the graph demonstrating a 6x difference. The images in the supplement are also highly overexposed for some of the blots. How do you know that these bands are specific?

2) Some of the echo parameters are off such as cardiac output which is double what is expected in a mouse and wall thickness are significantly smaller. EDV and ESV also seem very small. It is not clear if these measurements were taken from both long and short axis measurements at midpapillary?

3) It is unclear how the conclusion was mad that PgLPS was increasing RAS when there was no change in the AT receptor. While the downstream intracellular mechanism make sense, there is not a clear connection to the RAS system. It is possible that activation of RAS is not directly through Pg-LPS but another indirect pathway.

Minor:

1) Grammatic issue with sentence in abstract 3rd line: ‘have shown little beneficial effects of ACE inhibitors in animals with poor oral health, particularly periodontitis, has been shown in some experimental failing hearts.’ which has two verbs.

2) Page 7 in methods that is duplicated on last line.

Reviewer #3: It is a well-structured manuscript with great scientific and clinical impact, which is why I suggest its publication, however the following adjustments are required:

1. In the methodology, mention how the sample size of n: 6 of the animals per treatment group is determined.

2. Review the images of the western blot results, seven drafts of the bands and in some cases the relationship of those described and observed in the figure of bars with the increase in expression is not seen.

3. In the discussion they focus on describing the results obtained in this manuscript with what was previously obtained by the same authors, it is necessary to seek other scientific contributions described by other researchers.

6. PLOS authors have the option to publish the peer review history of their article (what does this mean?). If published, this will include your full peer review and any attached files.

Reviewer #1: No

Reviewer #2: No

Reviewer #3: No

---

## [Author Response · Author response to Decision Letter 0]

27 Jun 2023

Reviewer #1:

The present study aimed to evaluate the possible protective effects of oral Captopril in mice heart infected with Porphylomonas gingivalis LPS (LPS-Pg), i.p. The relationship between poor oral health and cardiovascular complications, such as heart failure and fibrosis, is widespread in the literature. The important role of RAS components in cardiac functions and structures is also known, especially when it comes to Ang-II peptide linked to AT1 receptor. 

Criticism-1:

It was the hypothesis of this study that there would be cardiovascular protection promoted by Captopril with regard to decrease in cardiac fibrosis caused by LPS-Pg, as well as an improvement in ejection fraction also in mice treated with LPS-Pg. However, the sample size calculation for carrying out this study was not presented, so there is no guarantee that the number of mice used reflect the reality of the results.

Response-1-(1): We asked Dr. Yasuharu Amitani, a statistician, who is included as a co-author in the revised manuscript, to examine the statistical validity in response to the comments of Reviewers #1 and #3. We calculated the total sample size of animals (α risk = 0.05, power (1-β) = 0.8) not only a priori (effect size (f) = 0.4) but also a posteriori with the effects size (f) derived post hoc [1] by means of G* Power version 3.1 (program, concept and design by Franz, Universitat Kiel, Germany; freely available Windows application software) [2] (S2 Data). In regard to the data with the statistically significant differences: Fig 1C, 3B, 3C, 4B, 4C, 5B-F and Table 1A (EF) and 1D (%FS) (S2 Data), the actual sample sizes are close to the required sample sizes (n = 20 - 28) because the a posteriori effect sizes (f) are much greater than the a priori size (f = 0.4) [1]. Therefore, the sample size used in this study might be sufficient to evaluate the significant differences.

We incorporated the following sentences in the method section of the revised manuscript (Page 13, Lines 4-8).

We calculated the required total sample size of animals (α risk = 0.05, power (1-β) = 0.8) not only a priori (effect size (f) = 0.4) but also a posterior with the effects size (f) derived from post hoc [1] by means of G* Power version 3.1 (program, concept and design by Franz, Universitat Kiel, Germany; freely available Windows application software) [2] (S2 Data).

Response-1-(2): However, in regard to the data without statistically significant differences: Fig 1B, 2A-D, 5A and Table 1B, 1C, 1E-M), the actual sample sizes are small, compared to the required sample size a posteriori (40-224). Although it is not possible to rule out the possibility that the lack of a detected difference is due to the relatively small sample size, we could not prepare enough animals within a reasonable timeframe. There are also ethical issues associated with the use of large number of animals. 

 Instead, we have incorporated a comment on this issue as a study limitation in the results section of the revised manuscript as shown below.

1) Page 14, Lines 13-17 

 ---chronic PG-LPS treatment with or without Cap under the experimental conditions used in this study had no significant effect on growth, food consumption or water consumption, although we cannot rule out the possibility that the apparent lack of difference was due to the relatively small number of animals used. 

2) Page 18, Lines 11-15 

 We thus examined the expression of AT1 receptors in the heart and found that the levels were similar among the four groups, although we cannot rule out the possibility that the apparent lack of difference was due to the relatively small number of animals used (Fig 5A).

Criticism-2:

In addition, the figure 6 concludes that the increase in oxidative stress with consequent increase in apoptosis and cardiac fibrosis, as well as the left ventricular ejection fraction worsening, is linked to Ang-II and AT1 receptors interaction. Bearing this in mind, the Captopril use could decrease the Ang-II availability and contribute to the improvement of cardiac function in mice infected with LPS-Pg. On the other hand, there is no mention in this study other possibilities of Ang-II formation by other enzymatic pathways besides the Converse Angiotensin enzyme (ACE) function. Elastase-2, for example, can cleave angiotensinogen directly into Ang-II, so an Ang-II dosage throughout the study is essential for figure 6 conclusion. 

Response-2: 

We examined serum levels of Ang II by means of ELISA. Serum Ang II levels were significantly increased in the PG-LPS group compared to the control (Control [n = 4] vs. PG-LPS [n = 4]: 88 ± 12 vs. 759 ± 624, P < 0.05 by one-way ANOVA followed by the Tukey-Kramer post hoc test), and the increase was significantly suppressed by Cap (PG-LPS [n = 4] vs. PG-LPS [n = 5]: 759 ± 624 vs. 62 ± 67, P < 0.05 by one-way ANOVA followed by the Tukey-Kramer post hoc test (Fig 1C).

These results suggest that activation of RAS might be mediated by PG-LPS treatment.

We incorporated the above sentences in the result section of the revised manuscript as shown below (Page 15, Line 12-Page 16, Line 3).

Effects of PG-LPS on serum Ang II levels

The renin-angiotensin system is thought to be involved in inflammatory processes such as periodontitis. However, its precise role is still unclear. Therefore, we next examined serum levels of Ang II in the four groups by means of ELISA. Serum Ang II levels were significantly increased in the PG-LPS group compared to the control (Control [n = 4] vs. PG-LPS [n = 4]: 88 ± 12 vs. 759 ± 624 pg/ml, P < 0.05 by one-way ANOVA followed by the Tukey-Kramer post hoc test), and the increase was significantly suppressed by Cap (PG-LPS [n = 4] vs. PG-LPS + Cap [n = 5]: 759 ± 624 vs. 62 ± 67 pg/ml, P < 0.05 by one-way ANOVA followed by the Tukey-Kramer post hoc test (Fig 1C).

 These results suggest that activation of RAS might be mediated by PG-LPS treatment.

Criticism-3:

Finally, it would be interesting and important to have access to complete Western Blot membranes photos of all target studies because in some cases, such as the making for BCL-2, AT-1 and XO, the bands shown are double or triple and this can configure low antibodies selectivity and efficiency. It is worth mentioning that the presented images blots are cropped. 

Response-3: 

We performed Western blotting of BCL-2, AT-1 and XO using immunoreaction enhancer solution (Can Get Signal; TOYOBO, #NKB-101, Osaka, Japan) to improve the sensitivity and specificity. We replaced the blots of BCL-2, AT-1 and XO with the new blots in the revised manuscript, and show the complete blots as supplementary data (see Fig 4C, Fig 5A, Fig 5D, S2 Fig of S1 Data, S3 Fig of S1 Data, and S6 Fig of S1 Data in the revised manuscript). 

We modified the sentences in the method section of the revised manuscript as shown below (Page 12, Lines 9-11).

The primary and secondary antibodies were diluted in Tris-buffered saline (pH 7.6) with 0.1% Tween 20 and 5% bovine serum albumin or immunoreaction enhancer solution (Can Get Signal; TOYOBO, #NKB-101, Osaka, Japan). 

Reviewer #2

The goal of this study was to investigate the effects of the ACE inhibitor captopril on Porphyromonas gingivalis lipopolysaccharide (PG-LPS) induced cardiac dysfunction in mice. The finding of this study are interesting and align with effects of oral disease state in the heart however, the conclusion that RAS is involved is a jump that is not substantiated by the data. In addition, there is clear issue with the data presented specifically with the echo and immunoblotting.

Criticism-1:

Immunoblots are not convincing as they are highly zoomed in and blurry. In some cases, the images do not visually look different despite the graph demonstrating a 6x difference. The images in the supplement are also highly overexposed for some of the blots. How do you know that these bands are specific?

Response-1: 

The reviewer may be referring especially to the Western blot data of XO (Fig 5D), i.e., the image does not visually look different despite the graph demonstrating a 6x difference. Please see our responses to the criticism-3 from the reviewer #1 and the criticism-2 from the reviewer #3. We have replaced the blots of XO with BCL-2 and AT-1 with new blots obtained using immunoreaction enhancer solution in the revised manuscript (see Fig 4C, Fig 5A, Fig 5D, S2 Fig of S1 Data, S3 Fig of S1 Data, and S6 Fig of S1 Data in the revised manuscript)

Criticism-2:

Some of the echo parameters are off such as cardiac output which is double what is expected in a mouse and wall thickness are significantly smaller. EDV and ESV also seem very small. It is not clear if these measurements were taken from both long and short axis measurements at midpapillary?

Response-2:

All LV dimensions are presented as the average of four consecutive selected beats. Heart rate (HR) was determined from the cardiac cycles recorded on the M-mode tracing, using at least three consecutive beats. The other parameters were calculated from M-mode-derived LV dimensions using the Teichholz formula [3]:

●EDV = (7 x LVIDd3/1000) / (2.4 + (LVIDd/10)) and ESV = (7 x LVIDs3/1000) / (2.4 + (LVIDd/10)) (mL)

EDV (mL): left ventricular end-diastolic volume

ESV (mL): left ventricular end-systolic volume

LVIDd (mm): left ventricular internal dimension at end-diastole

LVIDs (mm): left ventricular internal dimension at end-systole

● Stroke volume (SV) = EDV- ESV (mL)

● Cardiac output (CO) = HR x SV (ml/min)

● left ventricular ejection fraction (EF) = 100 x SV / EDV (%)

● left ventricular fractional shortening (%FS) = 100 x (LVIDd - LVIDs) / LVIDd (%)

All LV dimensions calculated using Teichholz formula in wild-type control (12-week-old C57BL/6 mice) shown in this study were consistent with those reported in previous studies by us [4] and another group [5]. 

We incorporated the above comments in the method section of the revised manuscript (Page 9, Lines 1-17).

Criticism-3:

It is unclear how the conclusion was mad that PGLPS was increasing RAS when these was no change in the AT receptor. While the downstream intracellular mechanism make sense, there is a clear connection to the RAS system. It is possible that activation of RAS is not directly through Pg-LPS but another indirect pathway.

Response-2: 

Please see our responses to the criticism-2 from reviewer #1.

Minor:

Criticism-1:

Grammatic issue with sentences in abstract 3rd line: ‘have shown little beneficial effects of ACE inhibitors in animals with poor oral health, particularly periodontitis, has been shown in some experimental failing hearts’ which has two verbs. 

Response-1:

Thank you. We have corrected it as shown below (Page 3, Lines 4-5).

---have shown little beneficial effects of ACE inhibitors in animals with oral health, particular periodontitis. In this study, we examined---

Crtiticism-2:

Page 7 in methods that is duplicated on last line.

Response-2:

Thank you. We corrected it as shown below (Page 7, Line 17-Page 8, Line 1)..

---this study is consistent with the circulating levels in patients with periodontitis, so that this model is not a sepsis model---

Reviewer 3:

It is well-structured manuscript with great scientific and clinical effects, which is why I suggest its publication, however the following adjustments are required.

Crtiticism-1:

In the methodology, mention how the sample size of n: 6 of the animals per treatment group is determined.

Response-1:

Please see our responses to the criticism-1 from reviewer #1.

Criticism-2:

Review the images of the western blot results, seven drafts of the bands and in some cases the relationship of those described and observed in the figure of bars with the increase in expression is not seen.

Response-2:

We think the reviewer was referring the Western blots of PKC-δ and XO (Fig 5B and D). Please see our responses to the criticism-3 from the reviewer #1 and criticism-1 from the reviewer #2. We replaced the blots of PKC-δ and XO with BCL-2 and AT-1 with new blots obtained using immunoreaction enhancer solution in the revised manuscript (see Fig 4C, Fig 5A, Fig 5B, Fig 5D, S2 Fig of S1 Data, S3 Fig of S1 data, S4 Fig of S1 Data and S6 Fig of S1 data in the revised manuscript). 

Criticism-3:

In the discussion they focus on describing the results obtained in this manuscript with what was previously obtained by the same authors, it is necessary to seek other scientific contributions described by other researchers.

Response-3:

The involvement of the RAS in myocardial fibrosis is evident in several pathological conditions such as hypertensive heart disease [6], congestive heart failure [7] and myocardial infarction [8]. It has been suggested that PD may be a contributory risk factor for CVD [9]. Although many studies have found an association between PD and CVD, the relationship is still controversial and is thought to require more investigation [10, 11]. To our knowledge, this study is the first to establish an association between PD and myocardial fibrosis and to afford evidence that RAS might be a key mediator for this association.

We incorporated the above sentences in the discussion portion of the revised manuscript (Page 27, Lines 1-8).

References

1. Cohen J. A power primer. Psychol Bull. 1992;112(1):155-9. 

2. Faul F, Erdfelder E, Buchner A, Lang AG. Statistical power analyses using G*Power 3.1: tests for correlation and regression analyses. Behav Res Methods. 2009;41(4):1149-60. 

3. Teichholz LE, Kreulen T, Herman MV, Gorlin R. Problems in echocardiographic volume determinations: echocardiographic-angiographic correlations in the presence of absence of asynergy. Am J Cardiol. 1976;37(1):7-11. 

4. Matsuo I, Kawamura N, Ohnuki Y, Suita K, Ishikawa M, Matsubara T, et al. Role of TLR4 signaling on Porphyromonas gingivalis LPS-induced cardiac dysfunction in mice. PLoS One. 2022;17(6):e0258823. 

5. Jones AA, Framnes-DeBoer SN, Shipp A, Arble DM. Caloric restriction prevents obesity- and intermittent hypoxia-induced cardiac remodeling in leptin-deficient ob/ob mice. Front Physiol. 2022;13:963762. 

6. Brilla CG, Pick R, Tan LB, Janicki JS, Weber KT. Remodeling of the rat right and left ventricles in experimental hypertension. Circ Res. 1990;67(6):1355-64. 

7. Weber KT, Brilla CG, Janicki JS. Myocardial fibrosis: functional significance and regulatory factors. Cardiovasc Res. 1993;27(3):341-8. Epub 1993/03/01. doi: 10.1093/cvr/27.3.341. PubMed PMID: 8490934.

8. Hanatani A, Yoshiyama M, Kim S, Omura T, Toda I, Akioka K, et al. Inhibition by angiotensin II type 1 receptor antagonist of cardiac phenotypic modulation after myocardial infarction. J Mol Cell Cardiol. 1995;27(9):1905-14. 

9. Humphrey LL, Fu R, Buckley DI, Freeman M, Helfand M. Periodontal disease and coronary heart disease incidence: a systematic review and meta-analysis. J Gen Intern Med. 2008;23(12):2079-86. 

10. Sanz M, Marco Del Castillo A, Jepsen S. Periodontitis and cardiovascular diseases: Consensus report. J Gen Intern Med 2020;47(3):268-88. 

11. Gonzalez-Navarro B, Pintó-Sala X, Corbella E, Jané-Salas E, Miedema MD, Yeboah J, et al. Associations between self-reported periodontal disease, assessed using a very short questionnaire, cardiovascular disease events and all-cause mortality in a contemporary multi-ethnic population: The Multi-Ethnic Study of Atherosclerosis (MESA). Atherosclerosis. 2018;278:110-6.

---

## [Decision Letter · Decision Letter 1]

20 Jul 2023

PONE-D-23-02675R1Oral angiotensin-converting enzyme inhibitor captopril protects the heart from Porphyromonas gingivalis LPS-induced cardiac dysfunction in micePLOS ONE

Dear Dr. Okumura,

Thank you for submitting your manuscript to PLOS ONE. After careful consideration, we feel that it has merit but does not fully meet PLOS ONE’s publication criteria as it currently stands. Therefore, we invite you to submit a revised version of the manuscript that addresses the points raised during the review process. Please submit your revised manuscript by Sep 03 2023 11:59PM. If you will need more time than this to complete your revisions, please reply to this message or contact the journal office at plosone@plos.org. Please include the following items when submitting your revised manuscript:A rebuttal letter that responds to each point raised by the academic editor and reviewer(s). You should upload this letter as a separate file labeled 'Response to Reviewers'.A marked-up copy of your manuscript that highlights changes made to the original version. You should upload this as a separate file labeled 'Revised Manuscript with Track Changes'.An unmarked version of your revised paper without tracked changes. You should upload this as a separate file labeled 'Manuscript'.If applicable, we recommend that you deposit your laboratory protocols in protocols.io to enhance the reproducibility of your results. Protocols.io assigns your protocol its own identifier (DOI) so that it can be cited independently in the future. For instructions see: https://journals.plos.org/plosone/s/submission-guidelines#loc-laboratory-protocols. Additionally, PLOS ONE offers an option for publishing peer-reviewed Lab Protocol articles, which describe protocols hosted on protocols.io. Read more information on sharing protocols at https://plos.org/protocols?utm_medium=editorial-email&utm_source=authorletters&utm_campaign=protocols.

We look forward to receiving your revised manuscript.

Kind regards,

Michael Bader

Academic Editor

PLOS ONE

Journal Requirements:

**Additional Editor Comments:**

Please take out the data on AT1A and Bcl2, the corresponding original blots are far from being convincing, since there are much stronger bands than the ones selected proving the non-specificity of the antibodies used. Also the P-PKCdelta and PCaMKII blots have a lot of additional bands, and should at least be discussed as possibly misleading.

Reviewers' comments:

Reviewer's Responses to Questions

**Comments to the Author**

1. If the authors have adequately addressed your comments raised in a previous round of review and you feel that this manuscript is now acceptable for publication, you may indicate that here to bypass the “Comments to the Author” section, enter your conflict of interest statement in the “Confidential to Editor” section, and submit your "Accept" recommendation.

Reviewer #1: All comments have been addressed

Reviewer #2: (No Response)

2. Is the manuscript technically sound, and do the data support the conclusions?

Reviewer #1: Yes

Reviewer #2: Partly

3. Has the statistical analysis been performed appropriately and rigorously? 

Reviewer #1: Yes

Reviewer #2: Yes

4. Have the authors made all data underlying the findings in their manuscript fully available?

Reviewer #1: Yes

Reviewer #2: Yes

5. Is the manuscript presented in an intelligible fashion and written in standard English?

Reviewer #1: Yes

Reviewer #2: Yes

6. Review Comments to the Author

Reviewer #1: This reviewer recognizes and congratulates the efforts and courage of the authors in returning appropriate answers to the questions initially asked during the reviewing process. Ang II dosage was fundamental to support the conclusions. All researchers who propose western blot technique to detect and quantify proteins, often face antibodies with poor sensitivity and specificity. Although some blots are not satisfactory, it is possible to observe the modulations described in the work, thus being adequate for publication. It is worth warning editors of excellent journals, such as PlosOne, in defining the Western blots quality to be accepted for publication so that manufacturers will make a greater effort to produce antibodies with more sensitivity and specificity.

Reviewer #2: While much of the comments have been addressed, all of the immunoblots are not satisfactory and seeing as much of the mechanism presented is based on these blots, the conclusions are not supported.

1) While attempts were made to improve the quality of the data, all immunoblots still are very blurry in the main figures. Most do not match the trend shown in the graphs.

2) Bcl2 and the PCaMKII blots look overexposed or poor antibody specificity as there are many bands with a stronger signal than the one selected for quantification.

3) All immunoblots in the supplement should be labeled in a clear manner so that all bands are labeled for which sample group it belongs to, not just the bands shown in the figure.

These technical issues significantly dampen my enthusiasm.

7. PLOS authors have the option to publish the peer review history of their article (what does this mean?). If published, this will include your full peer review and any attached files.

Reviewer #1: No

Reviewer #2: No

---

## [Author Response · Author response to Decision Letter 1]

22 Sep 2023

Additional Editor Comments:

Criticism-1

Please take out the data on AT1A and Bcl2, the corresponding original blots are far from being convincing, since there are much stronger bands than the ones selected proving the non-specificity of the antibodies used.

Response-1:

As requested, we have removed the data on AT1 receptor and Bcl-2 from the revised manuscript.

Criticism-2:

Also the PKCdelta and PCAMKII blots have a lot of additional bands, and should at least be discussed as possibly misleading.

Response-2:

Indeed, phospho-PKCδ (Tyr-311) blot and phospho-CaMKII (Thr-286) blot have many additional bands, so to select suitable bands for quantification, we carried out chronic infusion of Ang II dissolved in saline for 7 days at a dose of 1,500 ng/kg per min and isolated the heart after 7 days [1, 2]. Control mice received infusion of a comparable volume of saline. 

In the phospho-PKCδ (Tyr-311) blot, the density of a single band was increased by Ang II treatment (see S1A Fig of S1 Data). In the phospho-CaMKII (Thr-286) blot, the density of two adjacent bands was increased by Ang II treatment (see S8B Fig of S1 Data). Therefore, these bands were considered suitable for quantification. 

The bands we selected for quantification based on this experiment are consistent with previously reported bands of phospho-PKCδ (Tyr-311) [3] and phospho-CaMKII (Thr-286) [4]. 

We incorporated the following sentences in the method section of the revised manuscript (Page 12, Line 14-Page 13, Line 3). 

Note that the blots of phospho-PKCδ (Tyr-311) and phospho-CaMKII (Thr-286) had many additional bands. Therefore, to identify appropriate bands for quantification, chronic infusion of Ang II (#015-27911; FUJIFILM Wako Pure Chemical Corporation) dissolved in saline was performed for 7 days at a dose of 1,500 ng/kg/min via an osmotic mini-pump (Model 2001; AlZET, Cupertino, CA, USA) and the heart was isolated after 7 days [1, 2]. Control mice received infusion of a comparable volume of saline (S1Fig of S1 Data). Bands whose density was increased in the treated mice were selected for quantification. 

We also incorporated the following sentences in the results section of the revised manuscript. 

1) Page 18, Lines 13-15

We first confirmed the identify of the band phosphorylated by Ang II, because phospho-PKCδ (Tyr-311) shows many additional bands (S1A Fig of S1 Data).

2) Page 20, Lines 15-17

We first confirmed the identify of the bands phosphorylated by Ang II, because phospho-CaMKII (Tyr-286) shows many additional bands (S1B Fig of S1 Data).

Reviewer #1:

Criticism-1:

This reviewer recognizes and congratulates the efforts and courage of the authors in returning appropriate answers to the questions initially asked during the reviewing process. Ang II dosage was fundamental to support the conclusions. All researchers who propose western blot technique to detect and quantify proteins, often face antibodies with poor sensitivity and specificity. Although some blots are not satisfactory, it is possible to observe the modulations described in the work, thus being adequate for publication. It is worth warning editors of excellent journals, such as PloS One, in defining the Western blots quality to be accepted for publication so that manufactures will make a great effort to produce antibodies with more sensitivity and specificity.

Resposne-1:

Thank you for your valuable comments.

Reviewer #2:

Criticism-1: 

While attempts were made to improve the quality of the data, all immunoblots still are very blurry in the main figures. Most do not match the trend shown in the graphs.

Response-1:

Please refer to our responses to criticism-1 and criticism-2 from the editor. We have removed the data on AT1-AR and Bcl-2 from the revised manuscript as the editor requested. We have also added further explanation regarding multiple bands in the blots. 

Criticism-2:

Bcl2 and the PCaMKII blots look overexposed or poor antibody specificity as there are many bands with a stronger signals than the one selected for quantification.

Response-2:

The Bcl-2 blot was removed from the revised manuscript in response to criticism-1 from the editor (please refer to our response to criticism-1 from the editor). Regarding the phospho-CaMKII blot, we have added an explanation of the rationale for band selection (please see our response to criticism-2 from the editor). 

Criticism-3:

All immunoblots in the supplement should be labelled in a clear manner so that all bands are labelled for which sample group it belongs to, not just the bands shown in the figure.

Response-3:

We have labelled all immunoblots in the supplemental data as the reviewer required. 

References

1. Ren J, Yang M, Qi G, Zheng J, Jia L, Cheng J, et al. Proinflammatory protein CARD9 is essential for infiltration of monocytic fibroblast precursors and cardiac fibrosis caused by angiotensin II infusion. Am J Hypertens. 2011;24(6):701-707. 

2. Li Y, Zhang C, Wu Y, Han Y, Cui W, Jia L, et al. Interleukin-12p35 deletion promotes CD4 T-cell-dependent macrophage differentiation and enhances angiotensin II-Induced cardiac fibrosis. Arterioscler Thromb Vasc Biol. 2012;32(7):1662-74. 

3. Shan D, Guo S, Wu HK, Lv F, Jin L, Zhang M, et al. Cardiac ischemic preconditioning promotes MG53 secretion through H2O2-activated protein kinase C-δ signaling. Circulation. 2020;142(11):1077-1091. 

4. Ikeda S, Matsushima S, Okabe K, Ikeda M, Ishikita A, Tadokoro T, et al. Blockade of L-type Ca2+ channel attenuates doxorubicin-induced cardiomyopathy via suppression of CaMKII-NF-κB pathway. Sci Rep. 2019;9(1):9850.

---

## [Editor Report · Decision Letter 2]

25 Sep 2023

Oral angiotensin-converting enzyme inhibitor captopril protects the heart from Porphyromonas gingivalis LPS-induced cardiac dysfunction in mice

PONE-D-23-02675R2

Dear Dr. Okumura,

We’re pleased to inform you that your manuscript has been judged scientifically suitable for publication and will be formally accepted for publication once it meets all outstanding technical requirements.

Kind regards,

Michael Bader

Academic Editor

PLOS ONE
---

## [Editor Report · Acceptance letter]

7 Nov 2023

PONE-D-23-02675R2 

Oral angiotensin-converting enzyme inhibitor captopril protects the heart from *Porphyromonas gingivalis* LPS-induced cardiac dysfunction in mice 

Dear Dr. Okumura:

I'm pleased to inform you that your manuscript has been deemed suitable for publication in PLOS ONE. Congratulations! Your manuscript is now with our production department. 

Kind regards, 

on behalf of

Prof. Michael Bader 

Academic Editor

PLOS ONE